# C-, N-, S-, and P-Substrate Spectra in and the Impact of Abiotic Factors on Assessing the Biotechnological Potential of *Paracoccus pantotrophus*

Denise Bachmann [1], Upasana Pal [1], Julia A. Bockwoldt [1,2], Lena Schaffert [3], Robin Roentgen [4], Jochen Büchs [4], Jörn Kalinowski [3], Lars M. Blank [1] and Till Tiso [1,*]

[1] iAMB—Institute of Applied Microbiology, ABBt—Aachen Biology and Biotechnology, RWTH Aachen University, Worringerweg 1, D-52074 Aachen, Germany
[2] Chair of Microbiology, Technical University of Munich, D-85354 Freising, Germany
[3] Center for Biotechnology—CeBiTec, Bielefeld University, D-33615 Bielefeld, Germany
[4] AVT—Chair for Biochemical Engineering, RWTH Aachen University, D-52074 Aachen, Germany
* Correspondence: till.tiso@rwth-aachen.de; Tel.: +49-241-802-6649

**Abstract:** Modern biotechnology benefits from the introduction of novel chassis organisms in remedying the limitations of already-established strains. For this, *Paracoccus pantotrophus* was chosen for in-depth assessment. Its unique broad metabolism and robustness against abiotic stressors make this strain a well-suited chassis candidate. This study set out to comprehensively overview abiotic influences on the growth performance of five *P. pantotrophus* strains. These data can aid in assessing the suitability of this genus for chassis development by using the type strain as a preliminary model organism. The five *P. pantotrophus* strains DSM 2944$^T$, DSM 11072, DSM 11073, DSM 11104, and DSM 65 were investigated regarding their growth on various carbon sources and other nutrients. Our data show a high tolerance against osmotic pressure for the type strain with both salts and organic osmolytes. It was further observed that *P. pantotrophus* prefers organic acids over sugars. All of the tested strains were able to grow on short-chain alkanes, which would make *P. pantotrophus* a candidate for bioremediation and the upcycling of plastics. In conclusion, we were able to gain insights into several *P. pantotrophus* strains, which will aid in further introducing this species, or even another species from this genus, as a candidate for future biotechnological processes.

**Keywords:** *Paracoccus*; physiology; carbon source; growth temperature

## 1. Introduction

The times when biotechnology solely relied on a few chosen strains to do its bidding are slowly changing, as more novel chassis organisms are being discovered and developed. It is becoming clearer that there is huge potential in searching for desirable traits in the natural diversity of bacteria, which are still largely unknown despite the best research efforts; however, finding a promising new microbial candidate comes with a whole set of requirements that have to be fulfilled [1].

First, one has to define what exactly falls under the umbrella of "chassis organism". In an ideal world, biological engineers would be able to design a chassis organism bottom-up, working with a biological scaffold (for example, a minimal cell), which can be equipped with standardized components, to yield a completely characterized and fully controllable biological catalyst; however, in reality, chassis organisms are wild-type strains that show promising qualities in nature that can be expanded upon—for example, the ability to use cheap substrates, tolerance against harsh environments, or the production of interesting molecules.

In order to tame the wild type and make it suitable for a reliable and efficient biotechnological production process, not only is there a big knowledge gap that has to be closed [2],

but novel tools have to either be developed or adapted to be able to manipulate physiological properties. The first step to gain a better understanding is to analyze the metabolic limitations in depth, which not only include the testing of different electron donors and acceptors but also the growth efficiency under the influences of various other abiotic factors, such as pH, salinity, and temperature, or the presence of toxic compounds.

Under these pretenses, using a promising wild-type candidate to assess its suitability as a chassis organism, *Paracoccus pantotrophus* was chosen as the first strain of the diverse genus *Paracoccus*. The Gram-negative alphaproteobacterium *P. pantotrophus* was isolated from an effluent treatment system for the removal of sulfur nitrogen compounds [3], originally designated with the name *Thiosphaera pantotropha*. Only later was this organism assigned to the genus *Paracoccus* [4]. It is both facultative autotrophic and facultative anaerobic.

Due to its origin, *P. pantotrophus* is very-well-versed in surviving a highly variable environment and is able to perform aerobic denitrification as well as heterotrophic denitrification [5], which enable it to carry out anaerobic respiration instead of changing its metabolism to fermentation. It was the first alphaproteobacterium in which the entire sulfur oxidation (SOX) system was characterized [6,7]. Furthermore, under certain nutritional limitations, the production of polyhydroxyalkanoates (PHAs), storage molecules, can be induced [8].

The genus type strain *P. denitrificans*, a close relative of *P. pantotrophus*, is facultative autotrophic. It achieves this by reducing hydrogen and fixing $CO_2$ via the reductive pentose phosphate cycle. It possesses the entire tricarboxylic acid cycle (TCA) cycle and is not capable of using the glycolytic pathway, instead utilizing sugars via the Entner–Doudoroff or hexose monophosphate pathway, or a combination of the two [9]. While *P. pantotrophus* is usually not able to grow on methanol, unless it gains this ability through spontaneous mutation [10], *P. denitrificans* is able to use methanol aerobically (with the resulting $CO_2$ becoming fixed in return) and anaerobically while using nitrate or nitrite as terminal electron acceptors. It is interesting to note that *Paracoccus denitrificans* was the first bacterium in which the equivalent of the mitochondrial electron transport chain was discovered [9,11]. The three parts of the respiratory chain, complexes I, III, and IV, in mitochondria share the same redox groups with *P. denitrificans*, with the latter possessing fewer subunits. Compared to other bacteria they possess more subunits, which provide evolutionary insights into the development of mitochondrial respiration [12].

Both *P. pantotrophus* and *P. denitrificans* have been subjects of multiple studies. Many of them focused on bioremediation or wastewater treatment, especially considering efforts in denitrification [13–19] and the production of PHAs [8,20–22], among others.

Despite these promising characteristics and the research that went into investigating this strain, there is no established biotechnological application making use of it. A large variety of carbon sources could be used to produce biologically degradable bioplastics, with this strain serving as the biological catalyst. To help further understand this versatile organism, five *P. pantotrophus* strains were characterized in this study, with the type strain DSM 2944[T] in particular. To reach this goal several different approaches were used, from screening via BIOLOG technology, to Growth Profiler analyses, to classic growth experiments, to determine the nitrification and sulfur oxidation capabilities. This can be seen as a first step to guide this strain along the route toward an established chassis organism, where deep and detailed knowledge about said organism is needed.

## 2. Materials and Methods

### 2.1. Bacterial Strains and Growth Media

The five *Paracoccus pantotrophus* strains, DSM 2944[T] (T indicating the type strain), DSM 11072, DSM 11073, DSM 11104, and DSM 65, were obtained from the German Collection of Microorganisms and Cell Cultures (Braunschweig, Germany). The selection was based on the interesting metabolic capabilities of these strains [23], the already-known comparative data [4], and their promising genetic abilities [24]. The cells were grown at 30 °C in a Delft mineral salt medium (MSM) with a final concentration per liter of 3.88 g

of $K_2HPO_4$, 1.63 g of $NaH_2PO_4$, 2.00 g of $(NH_4)_2SO_4$, 0.1 g of $MgCl_2 \times 6\,H_2O$, 10 mg of EDTA, 2 mg of $ZnSO_4 \times 7\,H_2O$, 1 mg of $CaCl_2 \times 2\,H_2O$, 5 mg of $FeSO_4 \times 7\,H_2O$, 0.2 mg of $Na_2MoO_4 \times 2\,H_2O$, 0.2 mg of $CuSO_4 \times 5\,H_2O$, 0.4 mg of $CoCl_2 \times 6H_2O$, and 1 mg of $MnCl_2 \times 2\,H_2O$, supplemented with various carbon sources in different concentrations [25].

## 2.2. Phylogenetic Analysis

The computational phylogenetic classification of *Paracoccus* spp. was carried out holistically by using every fully available genome in the NCBI of every type strain in this genus. The assembly level is mentioned in the tree in order to provide a better understanding of the data quality implemented here. The tree is based on core genomes in order to ensure better statistical precision, as 16S-based approaches can give false results [26]. Multiple copies of ribosomal rRNA genes as well as intragenomic variability can also lead to problems [27]; thus, using the core genomes whenever possible for detailed phylogenetic analyses and maximized sequence support is advantageous [28].

The phylogenetic tree was constructed using the EDGAR (Efficient Database framework for comparative Genome Analyses using BLAST score Ratios) software, available via the following link: https://www.uni-giessen.de/fbz/fb08/Inst/bioinformatik/software/EDGAR (accessed on 9 January 2023) (Justus-Liebig-Universität, Gießen, Germany).

With EDGAR, a core genome can be calculated from selected genomes. The alignment of orthologous genes found in all genomes is performed via the multiple alignment tool MUSCLE [29]. Whole genomes can be compared in this way, and a more accurate relationship between the different species can be determined.

After the alignments are concatenated, a distance matrix is created by using the neighbor-joining method and a phylogenetic tree is constructed. This method is well-suited for large datasets due to its heuristic approach and good computational efficiency (https://www.uni-giessen.de/fbz/fb08/Inst/bioinformatik/software/EDGAR/Features/phyl_trees) (accessed on 9 January 2023).

The calculated tree is based on core genomes, that is, the set of orthologous genes found in all genomes instead of the 16S rRNA genes, as in the traditional approach, as the former provides superior results by combining multiple genes [28,30].

The tree is built by calculating the aforementioned orthologous gene set and aligning them with the MUSCLE (multiple sequence alignment with high accuracy and high throughput) tool [31], followed by concatenating them to one. Finally, the results are plotted in a distant matrix by using the neighbor-joining method [32] as implemented in the PHYLIP (phylogeny inference package) package; bootstrapping was performed for a qualitative analysis [33].

## 2.3. Temperature Profiling

To efficiently investigate the impact of different cultivation temperatures on microbial growth, a newly developed high-throughput temperature profiling system was applied, similar to the system described by Kunze et al. [34]. Compared to the published system, the new system is designed for 48-deepwell MTP FlowerPlates (m2p labs, Beckman Coulter, Baesweiler, Germany) in order to increase the maximum oxygen transfer capacity. With this high-throughput temperature profiling system, different cultivation temperatures can be studied in a single experiment. Different cultivation temperatures are achieved via a special aluminum block mounted on the MTP with radiators that fit into the well interspaces. By circulating hot and cold water through the edges of the block, a temperature gradient is adjusted, which is transferred into the MTP wells via the radiators. The optimal growth temperature was determined first in order to ensure relevant data in the following experiments.

## 2.4. Growth Experiments

Most of the carbon source screening was completed by using a Growth Profiler 960 (System Duetz, EnzyScreen BV, Heemstede, The Netherlands) at 30 °C and 250 rpm in

96-well MTP plates (CR1496dg: Polystyrene white square 96-half-deepwell microtiter plates), with sandwich covers (CR1396: Universal sandwich cover for 96-well MTPs) run for approx. 24 h. The online measured green values were then transformed into OD600, and subsequently the growth rates were calculated as described in [35].

Furthermore, cultures of 250 mL volume were grown in 500 mL Erlenmeyer flasks in Infors Multitron Pro shakers (Bottmingen, Switzerland) (30 °C, 200 rpm, 50 mm). The OD600 was measured with a photometer, and analytical samples were taken after indicated time periods.

### 2.5. Substrate Analysis

The degradation of the different substrates was analyzed for the detection of diauxic growth. The cultivation samples were centrifuged at 14,000 rpm, and the supernatant was stored at $-20$ °C until HPLC measurement. We used 5 mM sulfuric acid as a solvent at a flow rate of 0.4 mL/min. Samples cooled to 5 °C were injected with a volume of 20 μL into the HPLC column, BF-Serie, Metab-AAC, $300 \times 7.8$ mm (Isera GmbH, Düren, Germany), with a fitting precolumn at 40 °C. The detection of the substance was possible with a refraction indicator (RI) detector 2300 (KNAUER Wissenschaftliche Geräte GmbH, Berlin, Deutschland), as well as an RT and UV detector.

### 2.6. Nitrogen Analysis

The detection of nitrate and nitrite ions was performed using Lunges reagent. The generated $N_2$ gas was collected in inverted Durham tubes placed inside inoculum-containing Hungate tubes after 72 h of anaerobic incubation.

### 2.7. BIOLOG

Two types of phenotype microarrays (BIOLOG Inc.) were utilized in this study. The Gen III MicroPlate assay was performed by the DSMZ (Braunschweig, Germany), while the phenotype microarrays (PM1–PM20 [LS2], manufacturer's protocol using redox dye mix G) were analyzed at CeBiTec (Universität Bielefeld, Germany).

Each of these assays is based on respiration via the coloration of the media via the reaction of tetrazolium dye with NADH (an indicator for cell respiration). The coloration of the dye was then recorded over time by the OmniLog system (Mode 71000 Serial#406), and kinetic data were plotted to compare the influences of the different treatments by the manufacturer's software (Kinetik Analysis, Biolog and Omnilog 2.3). The standard carbon source for testing other nutrients, as well as abiotic factors, was glucose. For a more detailed overview of the procedure, refer to [36]. Subsequently, the data were analyzed via the opm package for R from Vaals [37] to obtain the area under a curve (AUC) values of each experiment in order to gain comparable values for respiration activity. To normalize the data, the difference between the AUC of the negative control, which contained no carbon source, and the experiment was used to obtain the ΔAUC.

## 3. Results

### 3.1. Detailed Phylogenic Analysis Places Paracoccus pantotrophus in a Distinct Branch in the Genus

To understand and establish the phylogenetic connection within the *Paracoccus* genus, 35 type strain genomes were selected.

The phylogenetic tree in Figure 1 shows that all *P. pantotrophus* genomes are grouped. *P. denitrificans* is closely related to *P. pantotrophus*, and this tree shows that the latter is indeed a descendant of the former. The other strains included in this tree do not show a clear grouping based on habitat, substrate, or product spectrum; however, some of the marine isolates seem to be more closely related to the out-group compared to the terrestrial isolates. The bootstrap values point to a diverse distribution inside the genus, with only a few strains being closely related.

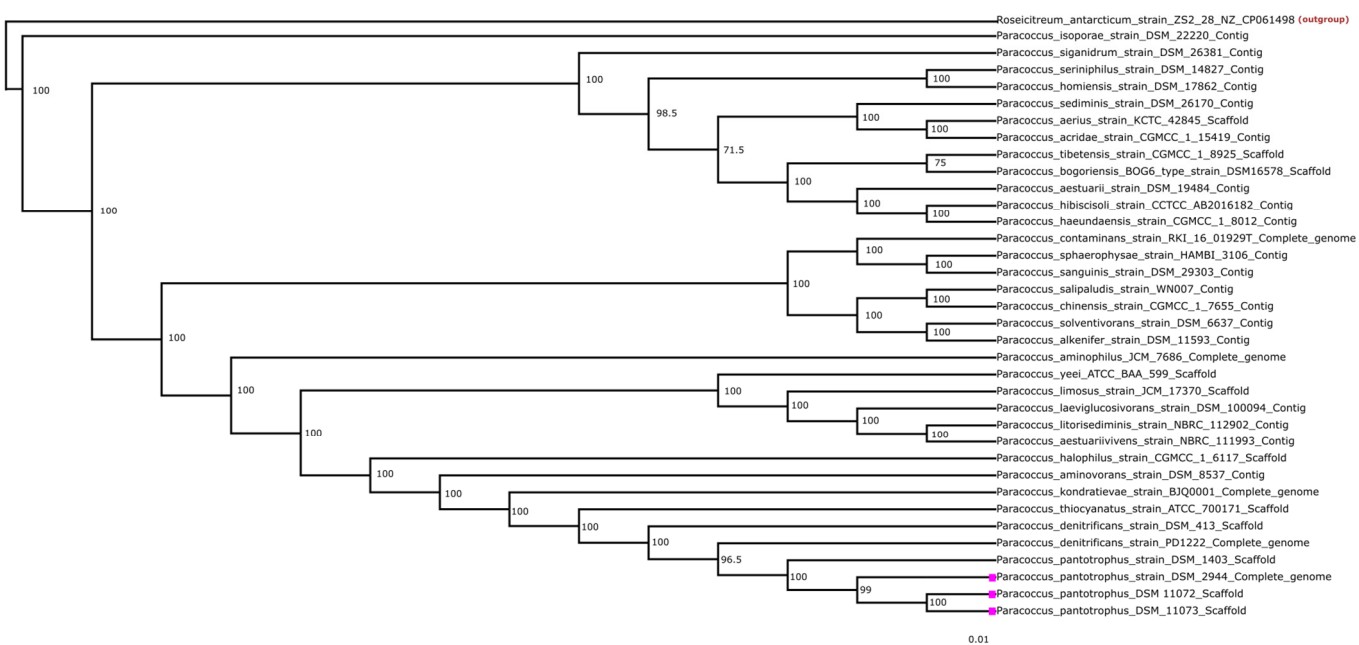

**Figure 1.** Rectangular phylogenetic tree of *Paracoccus*, comprising 35 *Paracoccus* genomes. A dataset of 992 genes per genome, 38,688 in total, was used. All genomes are annotated with genus, species, and assembly level, providing a holistic overview of the genome quality under study. The dataset has 361,894 amino acid residues and 14,113,866 in total. For the out-group and the starting point of the calculation, the genome of the bacterial strain *Roseicitreum antarcticum* ZS2-28 was chosen due to the high similarity to the *Paracoccus* genome (94.35% 16S identity). The genomes marked with pink boxes highlight the three strains used in this study and show the close phylogenetic similarity. Strains DSM 11104 and DSM 65 are not shown due to missing genomic data. The tree topology was validated using 100 bootstrapping with 200 iterations, as implemented in the Phylip package. Most branches showed 100% bootstrap support; the values are displayed in the image. The scale at the bottom depicts evolutionary distance.

### 3.2. Temperature Profiling Shows That P. pantotrophus Exhibits Mesophilic Growth Behavior

During preliminary experiments with the type strain, it could be determined that the optimal growth temperature is between 35 and 40 °C. Lower temperatures caused a delay until the stationary phase was reached and, concurrently, a lower growth rate. Ultimately, the same cell density was reached as the cultures grown during optimal conditions, while higher temperatures caused a lower final density.

After repetition with all of the *P. pantotrophus* strains (Figure S1), a similar pattern for every strain could be seen; however, some minor strain-specific differences emerged as well.

In the original description of the type strain [3], similar results were shown, with a possible temperature range of 15–42 °C and an optimum at 37 °C. With the data in this study, it could be proven that this is also true for the other four *P. pantotrophus* strains, with the additional information that the strains are more susceptible to high than to low temperatures, with the exception of DSM 2944$^T$, which showed the opposite trend (Figure 2). As the growth rates were most similar at 30 °C and the cultures grew with only a little delay, this temperature was chosen in further experiments for better performance comparisons.

### 3.3. High Metabolic Versatility of Investigated Strains

As *P. pantotrophus* is known to grow on a wide range of nutrients, the diversity of nutrient utilization inside this species with the five chosen strains was determined.

All investigated strains were extensively tested via BIOLOG microarrays, which measured respiratory activity in order to obtain an overview of the variability of their growth characteristics.

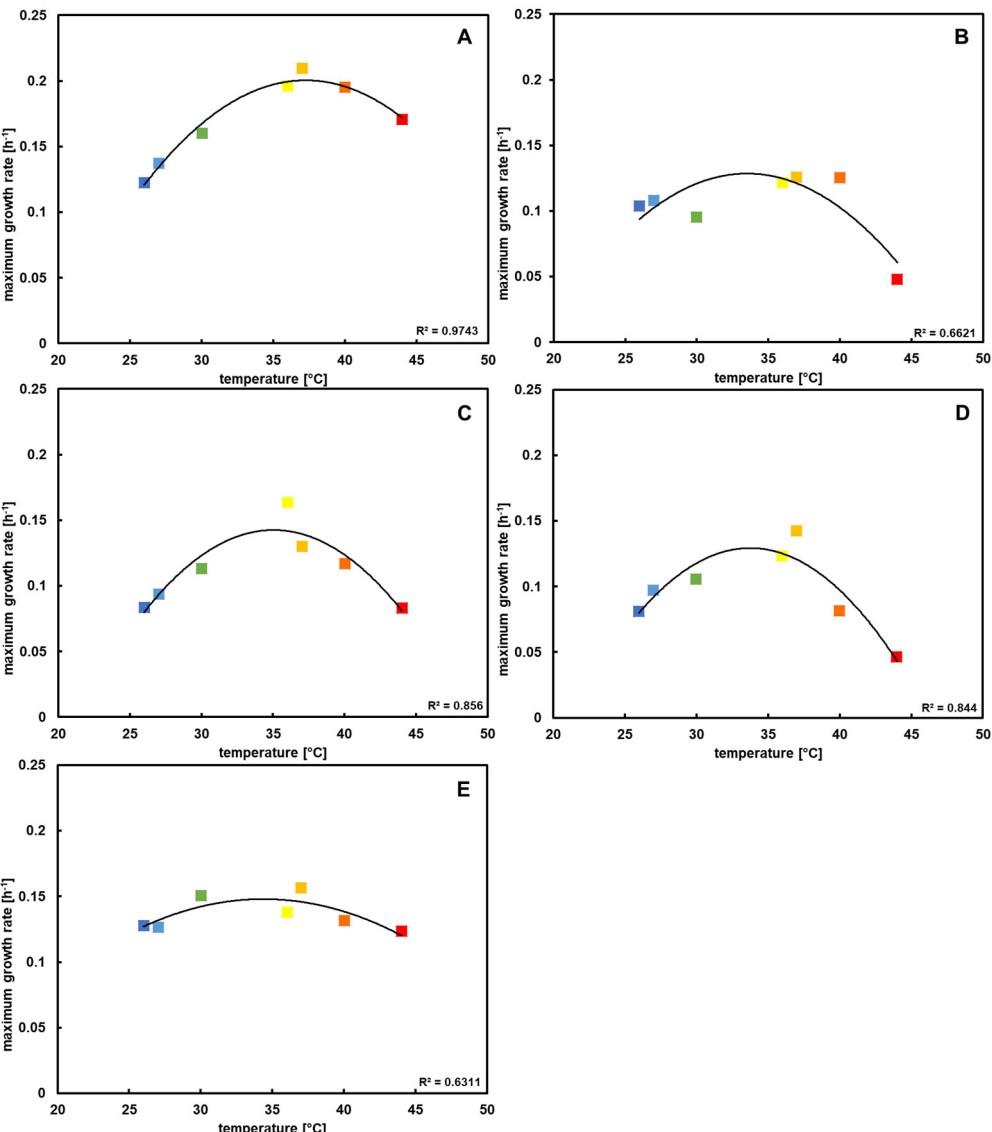

**Figure 2.** Maximum growth rates retrieved from the temperature block experiment after Kunze, Lattermann [34] plotted against growth temperature. All datasets correspond to a polynomal curve, with each maximum around mesophilic temperatures. (**A**) DSM 2944[T], (**B**) DSM 11072, (**C**) DSM 11073, (**D**) DSM 11104, and (**E**) DSM 65. The colors of the squares indicate the growth temperature.

In general, all strains showed a similar substrate utilization pattern (Figure 3). Different butyric acid compounds, disaccharides, and organic acids showed strong respiration for every strain. The type strain had a wider range of substrates that caused high respiration activity, in addition to being able to use D-malic acid, D-sorbitol, α-hydroxybutyric acid, D-fructose, L-alanine, and L-lactic acid better than the other strains. It stood out that L-galactonic acid-γ-lactone could be used much better by the type strain in comparison to the others, while DSM 11072 and 11073 were able to utilize N-acetyl-β-D-mannosamine considerably better than the rest of the tested strains.

Moderately strong respiration could be found for half of the tested compounds, with glycerol providing mixed results. Amino acids showed low respiration in general.

As for other non-nutrient factors, it is worthwhile noting that lithium chloride, a common bacterial inhibitor, had no notable negative effect on the vitality of all strains. Furthermore, none of the strains aside from DSM 11104 seemed to be significantly inhibited by 8% NaCl in the medium.

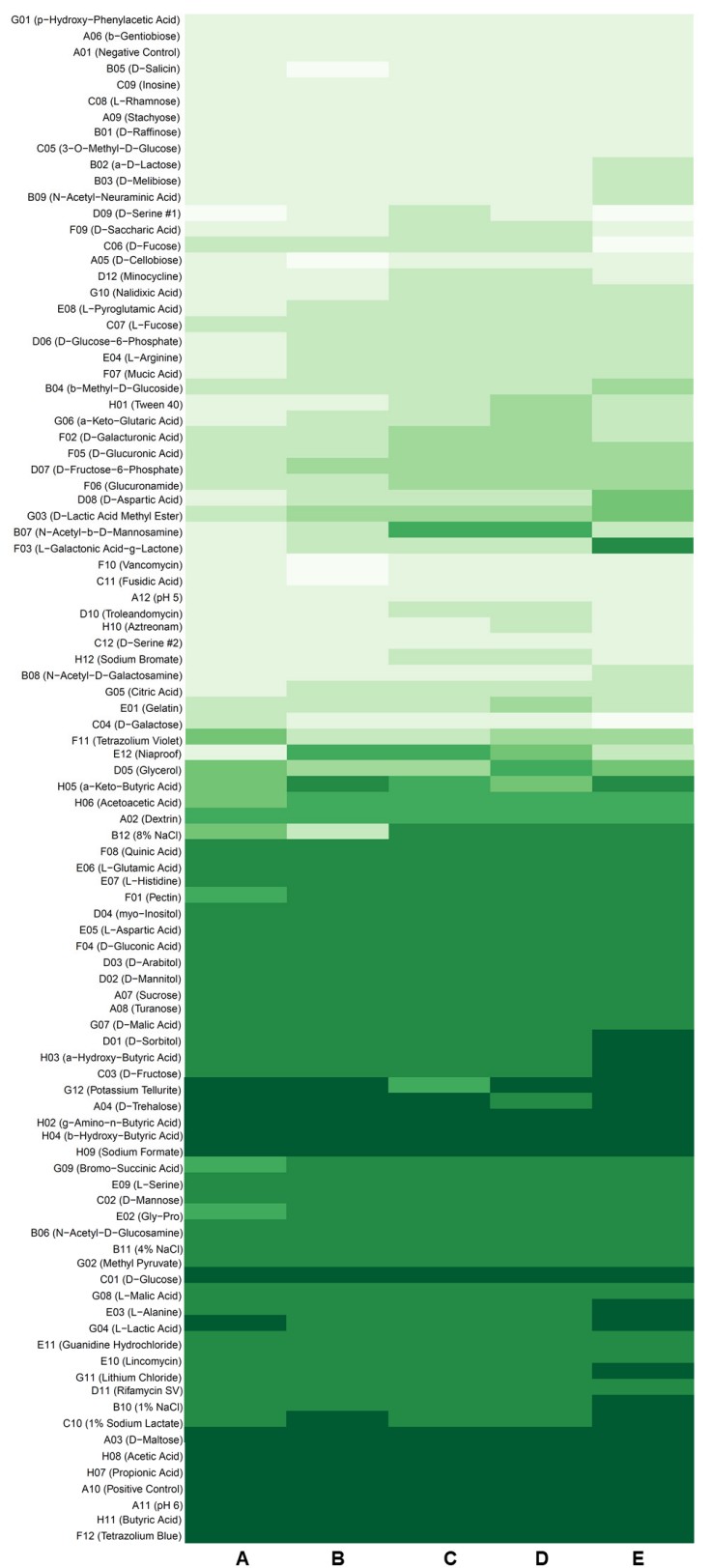
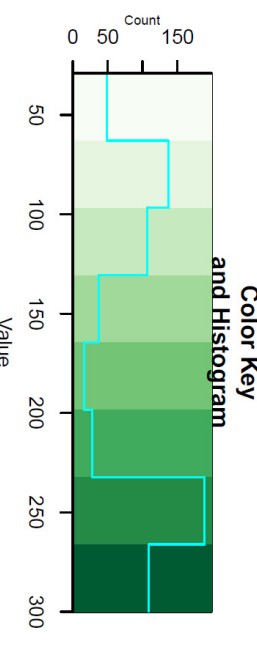

**Figure 3.** Heat map of BIOLOG Gen III MicroPlate analysis. This assay employs tetrazolium dye to measure the respiration of a standardized cell suspension, where stronger respiration leads to higher coloration. Darker shades of green correspond to stronger measured respiration. The cyan line in the legend shows the count for each shade in the dataset. Each tested compound was added in standardized amounts to a defined minimal medium. (**A**) DSM 65, (**B**) DSM 11104, (**C**) DSM 11073, (**D**) DSM 11072, and (**E**) DSM 2944[T].

None of the strains were able to thrive at a pH below 5, while at a pH of 6 respiration was observed. Certain tested antibiotics, such as aztreonam, showed strong inhibition in all strains, as expected.

This overview provides the basis for further investigations into metabolic characteristics as well as the influence of abiotic factors on growth.

For a further in-depth analysis of its metabolic potential, the type strain underwent an additional detailed BIOLOG investigation, and all of the strains were furthermore tested regarding their carbon, nitrogen, and sulfur usage, as well as the influence of abiotic factors such as osmotic stress or solvent sensitivity.

*3.4. Carbon Sources*

To determine the opportunities for *P. pantotrophus* in biotechnological applications, different methods were employed in order to gain a holistic picture of the catabolic capabilities of the five investigated strains. The following section showcases the detailed results of different studies.

3.4.1. Growth Efficiency on Selected Substrates Gives More Insight into *P. pantotrophus* Metabolic Variability

To validate the findings of the compact BIOLOG assay of all of the strains, growth data were obtained for every strain, as was respiration data for the type strain.

The growth of the five *P. pantotrophus* strains was investigated for a wide range of carbon sources in a Growth Profiler. All of the investigated strains showed very similar substrate utilization (Table 1). In general, C6 sugars and disaccharides were utilized for growth. Some simple organic acids, such as lactate, propionate, succinate, and acetate, were utilized. Most of the sugar alcohols tested were able to be used as a carbon source for growth. The type strain DSM 2944$^T$ showed mixed results with pure and crude glycerol, while two strains did not show growth with crude glycerol at all. Aromatic compounds were poorly utilized, and growth only occurred with styrene as the sole carbon source (data not shown). Most of the alcohols tested could be used as a carbon source, aside from methanol. Notably, some cultures showed growth after prolonged incubation with methanol, hinting at substrate adaptation via random mutation [10]. Of the fatty acids tested, only butyric acid could be used for growth by all of the strains. Acetoin, a molecule found in fermenting bacteria, could be utilized as carbon source by all of the strains. A wide variety of amino acids were used by all of the strains as carbon sources, whereas the type strain showed the highest growth rates of all with glutamate and histidine. Glutamate in general showed the best influence on growth, showing relatively high rates, except for DSM 11072.

These findings are comparable to the data from previous BIOLOG experiments, as they show growth on sugars and organic acids being prevalent in these strains. DSM 2944$^T$ was not able to grow on malic acid, even if respiration could be measured. All of the strains showed growth with sorbitol. Glycerol showed good respiration with the type strain, but it could not prove itself to be a good carbon source for growth for this organism. Lactate showed good results in terms of both growth and respiration.

This screening showed an impressive range of possible substrates, which prompted interest in a deeper understanding of the quality of utilization, as well as other growth characteristics which were achieved via the BIOLOG microarrays in the following chapters.

3.4.2. Respiration Analysis Highlights the Type Strain's Ability to Utilize a Wide Range of Organic Sources as Electron Donors

With the data from the full BIOLOG phenotype microarray experiments, the utilization of 190 carbon sources by the type strain could be determined. Of all of the substances, 63 (33%) showed a positive influence on respiration activity compared to the negative control (Table S1). The best respiration could be found with organic acids and amino acids. Saccharides showed slightly lower values. Sugar derivatives were not used well, aside from D-mannitol. Many unusual substances, including D-glucosamine, M-inositol, pectin,

methyl pyruvate, and 3-O-β-D-galactopyranosyl-D-arabinose, showed comparatively high values as well.

Considering the similar percentages of positive tested sources of each category, it becomes clear that no substrate class is obviously preferred over the other. This knowledge opens up the possibility to find substrates for co-feeding experiments, supporting the metabolism during growth in large volumes, as well as tailoring certain product compositions through the use of different nutrient concentrations.

**Table 1.** Overview of maximum growth rates ($h^{-1}$) for the five investigated strains with different substances as sole carbon sources. All of the experiments were performed in triplicates in 96-well plates with an MSM medium in a Growth Profiler. Online green value data were recorded and values were transformed into OD values to automatically calculate growth rates via an in-house MATLAB script. nd = no exponential growth detected.

| | | DSM 2944$^T$ | DSM 11072 | DSM 11073 | DSM 11104 | DSM 65 |
|---|---|---|---|---|---|---|
| Saccharides | D-Ribose | nd | nd | nd | nd | nd |
| | D-Xylose | nd | nd | nd | nd | nd |
| | L-Rhamnose | nd | nd | nd | nd | nd |
| | L-Arabinose | nd | nd | nd | nd | nd |
| | D-Glucose | 0.19 ± 0.00 | 0.46 ± 0.02 | 0.35 ± 0.01 | 0.25 ± 0.01 | 0.21 ± 0.00 |
| | D-Fructose | 0.42 ± 0.01 | 0.46 ± 0.01 | 0.41 ± 0.01 | 0.43 ± 0.02 | 0.44 ± 0.02 |
| | D-Galactose | nd | nd | nd | nd | nd |
| | D-Mannose | 0.07 ± 0.00 | 0.05 ± 0.00 | 0.05 ± 0.00 | 0.07 ± 0.00 | 0.07 ± 0.00 |
| | D-Trehalose | 0.41 ± 0.01 | 0.40 ± 0.01 | 0.18 ± 0.01 | 0.80 ± 0.05 | 0.38 ± 0.01 |
| | D-Saccharose | 0.30 ± 0.01 | 0.45 ± 0.01 | 0.29 ± 0.02 | 0.24 ± 0.01 | 0.17 ± 0.00 |
| | D-Lactose | nd | nd | nd | nd | nd |
| | D-Cellobiose | nd | nd | nd | nd | nd |
| | D-Maltose | 0.39 ± 0.01 | 0.45 ± 0.02 | 0.27 ± 0.01 | 0.30 ± 0.02 | 1.18 ± 0.1 |
| | Starch | nd | nd | nd | nd | nd |
| Organic acids | Lactate | 0.62 ± 0.02 | 0.48 ± 0.03 | 0.69 ± 0.05 | 0.71 ± 0.04 | 0.63 ± 0.03 |
| | Propionate | nd | 0.10 ± 0.00 | nd | 0.09 ± 0.00 | 0.06 ± 0.00 |
| | Succinate | 0.66 ± 0.03 | 0.54 ± 0.02 | 0.60 ± 0.02 | 0.77 ± 0.04 | 0.73 ± 0.05 |
| | Acetate | 0.78 ± 0.04 | 0.53 ± 0.03 | 0.70 ± 0.04 | 0.71 ± 0.03 | 0.64 ± 0.03 |
| | Malate | nd | nd | nd | nd | nd |
| | Adipic acid | nd | nd | nd | 0.02 ± 0.00 | nd |
| | Oxalate | nd | nd | nd | nd | nd |
| | Glycolate | nd | nd | nd | nd | nd |
| | Glyoxylate | nd | nd | nd | 0.23 ± 0.01 | nd |
| | Itaconic acid | nd | 0.19 ± 0.01 | nd | nd | nd |
| | Formate | 0.14 ± 0.01 | 0.47 ± 0.03 | 0.14 ± 0.01 | 0.08 ± 0.00 | 0.21 ± 0.01 |
| | D-Galacturonic acid | nd | nd | nd | nd | 0.13 ± 0.00 |
| | Glucuronic acid | nd | nd | nd | nd | nd |
| | Terephthalic acid | nd | nd | nd | nd | nd |
| | Citrate | nd | nd | nd | nd | nd |
| Sugar alcohols | Mannitol | 0.54 ± 0.04 | 0.57 ± 0.04 | 0.42 ± 0.03 | 0.32 ± 0.02 | 0.35 ± 0.02 |
| | Sorbitol | 0.40 ± 0.02 | 0.41 ± 0.02 | 0.33 ± 0.02 | 0.36 ± 0.02 | 0.27 ± 0.01 |
| | Xylitol | 0.34 ± 0.02 | 0.21 ± 0.01 | 0.44 ± 0.03 | 0.14 ± 0.00 | 0.14 ± 0.00 |
| | Adonitol | 0.44 ± 0.02 | 0.47 ± 0.04 | 0.50 ± 0.02 | 0.56 ± 0.04 | 0.37 ± 0.02 |
| | Erythritol | nd | nd | nd | nd | 0.45 ± 0.04 |
| | Acetoin | 0.44 ± 0.02 | 0.43 ± 0.02 | 0.40 ± 0.02 | 0.42 ± 0.02 | 0.25 ± 0.00 |
| | Glycerol | nd | 0.05 ± 0.00 | nd | 0.13 ± 0.01 | 0.03 ± 0.00 |
| | Crude glycerol | nd | nd | nd | nd | 0.13 ± 0.00 |
| | Ethylene glycol | 0.19 ± 0.00 | 0.08 ± 0.00 | 0.07 ± 0.00 | 0.12 ± 0.01 | 0.05 ± 0.00 |
| | Na-Butyrate | 0.36 ± 0.02 | 0.29 ± 0.01 | 0.34 ± 0.01 | 0.32 ± 0.01 | 0.15 ± 0.00 |
| | Decanoid acid | 0.28 ± 0.01 | 0.25 ± 0.01 | nd | nd | 0.10 ± 0.00 |
| | Hexadecanoic acid | nd | nd | nd | nd | nd |
| | Hexadecanoic acid | nd | nd | nd | nd | nd |
| | Stearic acid | nd | nd | nd | nd | nd |

**Table 1.** *Cont.*

| | | DSM 2944[T] | DSM 11072 | DSM 11073 | DSM 11104 | DSM 65 |
|---|---|---|---|---|---|---|
| **Amino acids** | L-Alanine | 0.34 ± 0.01 | 0.33 ± 0.01 | 0.19 ± 0.01 | 0.32 ± 0.01 | 0.31 ± 0.02 |
| | L-Arginine | nd | nd | nd | nd | nd |
| | L-Asparagine | 0.53 ± 0.04 | 0.15 ± 0.00 | 0.15 ± 0.00 | 0.05 ± 0.00 | 0.13 ± 0.00 |
| | L-Aspartic acid | 0.37 ± 0.01 | 0.34 ± 0.01 | 0.19 ± 0.01 | 0.29 ± 0.01 | 0.18 ± 0.01 |
| | L-Cysteine hydrochloride | nd | nd | nd | nd | nd |
| | L-Glutamine | 0.07 ± 0.00 | 0.03 ± 0.00 | 0.07 ± 0.00 | 0.01 ± 0.00 | 0.05 ± 0.00 |
| | L-Monosodium glutamate | 0.75 ± 0.04 | nd | 0.50 ± 0.03 | 0.16 ± 0.01 | 0.61 ± 0.05 |
| | Glycine | nd | 0.07 ± 0.00 | 0.04 ± 0.00 | nd | 0.05 ± 0.00 |
| | L-Histidine | 0.61 ± 0.03 | 0.22 ± 0.01 | 0.20 ± 0.01 | 0.32 ± 0.01 | 0.49 ± 0.03 |
| | L-Isoleucine | 0.18 ± 0.01 | 0.11 ± 0.00 | 0.12 ± 0.01 | 0.07 ± 0.00 | 0.11 ± 0.00 |
| | L-Leucine | 0.20 ± 0.01 | 0.14 ± 0.00 | 0.12 ± 0.00 | 0.06 ± 0.00 | 0.10 ± 0.00 |
| | L-Lysine | nd | nd | nd | nd | nd |
| | L-Methionine | nd | nd | nd | nd | nd |
| | L-Phenylalanine | 0.12 ± 0.01 | 0.08 ± 0.00 | nd | nd | 0.08 ± 0.00 |
| | L-Proline | 0.25 ± 0.01 | 0.25 ± 0.01 | 0.19 ± 0.01 | 0.07 ± 0.01 | 0.27 ± 0.01 |
| | L-Serine | 0.16 ± 0.00 | 0.13 ± 0.00 | 0.07 ± 0.00 | 0.10 ± 0.01 | 0.09 ± 0.00 |
| | L-Threonine | nd | nd | nd | nd | nd |
| | L-Tryptophane | nd | nd | nd | nd | nd |
| | L-Tyrosine | nd | nd | nd | nd | nd |
| | L-Valine | nd | 0.07 ± 0.00 | nd | nd | 0.13 ± 0.01 |
| **Others** | Ethanol | 0.20 ± 0.00 | 0.13 ± 0.00 | 0.15 ± 0.01 | 0.22 ± 0.00 | 0.62 ± 0.04 |
| | Gluconate | 0.41 ± 0.01 | 0.46 ± 0.02 | 0.49 ± 0.03 | 0.43 ± 0.02 | 0.25 ± 0.01 |
| | 1,4-Butanediol | nd | nd | 0.10 ± 0.00 | nd | 0.05 ± 0.00 |
| | 4,4-Diaminodiphenylmethan | nd | nd | nd | nd | nd |
| | Sunflower oil | nd | nd | nd | nd | 0.26 ± 0.02 |

The T indicates the type strain.

As it is known that *Paracoccus* sp. performs diauxic growth in the presence of multiple carbon sources, more tests were performed to determine the exact growth kinetics.

### 3.4.3. Diauxic Growth Behavior Shows That *P. pantotrophus* Favors Organic Acids over Glucose

The *P. pantotrophus* type strain DSM 2944[T] was tested for diauxic growth on a mixture of selected organic acids (succinate, lactate, and acetate) and fructose in combination with glucose to determine preferred carbon sources (Figure 4). The acids were chosen based on the high growth rates and sustainable sourcing of the compound.

The growth rates for all of the substrate combinations are relatively similar, with succinate showing the lowest value and lactate showing the highest value (Table 2).

**Table 2.** Maximum growth rates recorded in the diauxic growth experiment, using optical density as a proxy for cell number. N = 2.

| Substrate | Growth Rate (h$^{-1}$) |
|---|---|
| Glu-Fruc | 0.33 ± 0.02 |
| Glu-Ace | 0.36 ± 0.04 |
| Glu-Suc | 0.37 ± 0.03 |
| Glu-Lac | 0.34 ± 0.02 |

All of the organic acids showed very quick depletion during the exponential growth phase, while the glucose concentration stayed constant in each instant. The only time glucose was depleted was with fructose co-feeding, where both sugars show similar depletion during exponential growth.

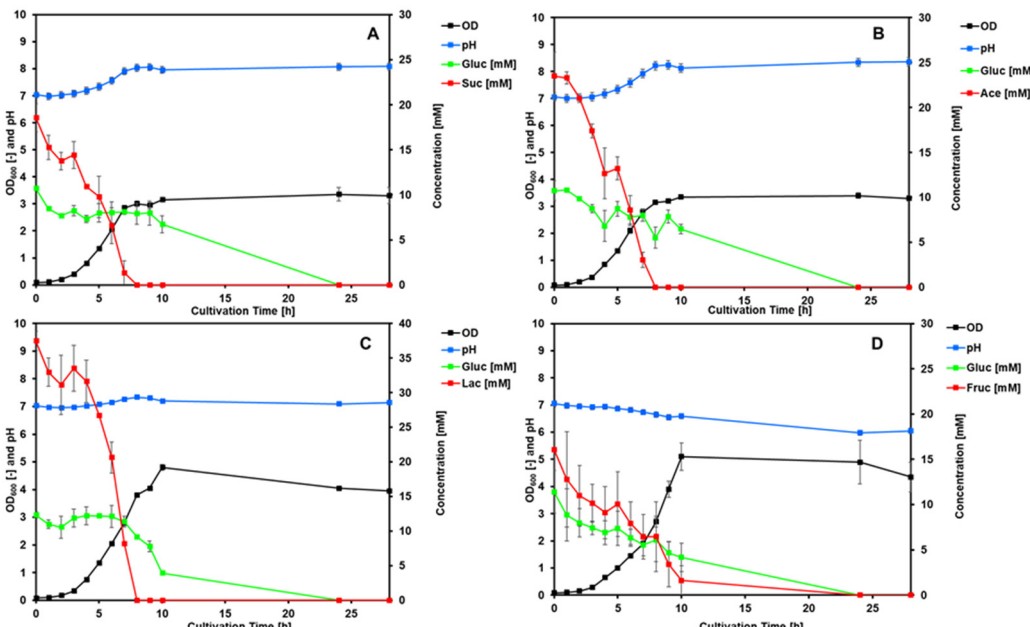

**Figure 4.** Comparison of substrate utilization by the type strain DSM 2944$^T$. Glucose was co-fed with different organic acids to determine diauxic growth, with fructose as a control. The glucose start concentration was chosen to be 10 mM, while the start concentrations of the second carbon sources were 2–3 times higher. The error bars represent the standard deviation of duplicates. The four graphs show the co-feeding of 10 mM glucose with (**A**) 20 mM succinate; (**B**) 25 mM acetate; (**C**) 40 mM lactate; and (**D**) 20 mM fructose.

Compared to the BIOLOG data, where the highest respiration could be found with organic acids, these data confirm the tendency of *P. pantotrophus* to utilize these carbon sources more readily and show that it can adapt its metabolic pathways to optimize growth rates depending on the available substrates.

The depletion of the first carbon source happened at different times (succinate: 9 h, fructose: 24 h, lactate: 8 h, and acetate: 8 h), after which growth occurred on glucose. All cultures entered the stationary phase of growth only after both carbon sources were depleted. This supports the findings of the microarray screening, where organic acids were found to be used most efficiently. While carbon sources and electron donors are very important when considering metabolic activity, other essential nutrients and their effects on the organism need to be considered as well. The other major component needed for biomass synthesis, and therefore for growth, is nitrogen.

### 3.5. Nitrogen Sources

*Paracoccus pantotrophus* is known to be able to perform heterotrophic nitrification and aerobic denitrification, for which all strains were tested, as was their ability to use all of the 21 proteinogenic amino acids as nitrogen sources. Analyses via BIOLOG microarrays were conducted to complement the results from the cultivation experiments.

3.5.1. All Strains Show High Variation in Growth Dynamics on Proteinogenic Amino Acids

To test the efficiency of essential amino acids as nitrogen sources, experiments with a Growth Profiler were conducted.

The precultures for all of the cultivations were grown on MSM with 20 mM of glucose. The ammonium sulfate in MSM was replaced with different amino acids for the main experiment.

The highest growth rates were reached by the type strain with L-histidine and L-alanine. L-cysteine hydrochloride caused slow growth with all of the strains (Table 3). DSM 11073 showed the highest growth rates of all, having no value lower than 0.13 h$^{-1}$,

barring L-cysteine hydrochloride ($0.06 \, h^{-1}$). It can be assumed that these amino acids also functioned as carbon sources for the strains, as stated in Table 1, where histidine showed a very good influence on the growth rates of all strains.

**Table 3.** Growth rates ($h^{-1}$) reached with amino acids as nitrogen sources. The experiments were performed in triplicates in 96-well plates at 30 °C with 20 mM glucose over the course of 24 h. Online green value data were recorded, and values were transformed into OD values to automatically calculate growth rates via an in-house MATLAB script. nd = no exponential growth detected.

| Amino Acid | DSM 2944$^T$ | DSM 11072 | DSM 11073 | DSM 11104 | DSM 65 |
|---|---|---|---|---|---|
| L-Alanine | $0.51 \pm 0.04$ | $0.29 \pm 0.01$ | $0.36 \pm 0.02$ | $0.26 \pm 0.00$ | $0.34 \pm 0.02$ |
| L-Arginine | $0.12 \pm 0.00$ | $0.08 \pm 0.00$ | $0.40 \pm 0.02$ | $0.16 \pm 0.01$ | $0.18 \pm 0.01$ |
| L-Asparagine | $0.39 \pm 0.02$ | $0.23 \pm 0.00$ | $0.28 \pm 0.01$ | $0.21 \pm 0.00$ | $0.27 \pm 0.01$ |
| L-Aspartic acid | $0.30 \pm 0.01$ | $0.15 \pm 0.00$ | $0.32 \pm 0.01$ | $0.31 \pm 0.00$ | $0.13 \pm 0.00$ |
| L-Cysteine hydrochloride | $0.05 \pm 0.00$ | $0.06 \pm 0.00$ | $0.06 \pm 0.00$ | $0.06 \pm 0.00$ | $0.06 \pm 0.00$ |
| L-Glutamine | $0.41 \pm 0.02$ | $0.25 \pm 0.01$ | $0.25 \pm 0.00$ | $0.21 \pm 0.00$ | $0.38 \pm 0.02$ |
| L-Monosodiumglutamate | $0.23 \pm 0.02$ | nd | $0.16 \pm 0.01$ | $0.11 \pm 0.00$ | $0.07 \pm 0.00$ |
| Glycine | $0.21 \pm 0.01$ | $0.11 \pm 0.00$ | nd | $0.33 \pm 0.01$ | $0.19 \pm 0.01$ |
| L-Histidine | $0.73 \pm 0.04$ | $0.34 \pm 0.01$ | $0.18 \pm 0.01$ | $0.30 \pm 0.00$ | $0.37 \pm 0.02$ |
| L-Isoleucine | $0.08 \pm 0.00$ | $0.05 \pm 0.00$ | $0.19 \pm 0.01$ | $0.16 \pm 0.01$ | $0.13 \pm 0.00$ |
| L-Leucine | $0.13 \pm 0.01$ | $0.10 \pm 0.00$ | $0.20 \pm 0.01$ | $0.22 \pm 0.01$ | $0.14 \pm 0.00$ |
| L-Lysine | $0.05 \pm 0.00$ | $0.08 \pm 0.00$ | $0.20 \pm 0.01$ | $0.29 \pm 0.02$ | $0.13 \pm 0.01$ |
| L-Methionine | $0.07 \pm 0.00$ | $0.06 \pm 0.00$ | $0.13 \pm 0.01$ | $0.19 \pm 0.01$ | $0.07 \pm 0.00$ |
| L-Phenylalanine | $0.06 \pm 0.00$ | $0.09 \pm 0.00$ | $0.15 \pm 0.01$ | $0.13 \pm 0.01$ | $0.13 \pm 0.01$ |
| L-Proline | $0.36 \pm 0.02$ | $0.23 \pm 0.01$ | $0.33 \pm 0.01$ | $0.31 \pm 0.01$ | $0.23 \pm 0.01$ |
| L-Serine | $0.42 \pm 0.02$ | $0.24 \pm 0.00$ | $0.36 \pm 0.02$ | $0.33 \pm 0.01$ | $0.21 \pm 0.00$ |
| L-Threonine | $0.06 \pm 0.00$ | $0.06 \pm 0.00$ | $0.17 \pm 0.01$ | $0.20 \pm 0.01$ | $0.06 \pm 0.00$ |
| L-Tryptophane | $0.08 \pm 0.01$ | $0.07 \pm 0.00$ | $0.20 \pm 0.01$ | $0.21 \pm 0.01$ | $0.22 \pm 0.01$ |
| L-Tyrosine | $0.10 \pm 0.00$ | $0.26 \pm 0.02$ | nd | $0.23 \pm 0.02$ | $0.17 \pm 0.01$ |
| L-Valine | $0.09 \pm 0.00$ | $0.08 \pm 0.00$ | $0.20 \pm 0.01$ | $0.21 \pm 0.00$ | $0.07 \pm 0.00$ |

The T indicates the type strain.

### 3.5.2. Denitrification Is Possible by All Investigated Strains

Due to the rising problem of eutrophication and the concurrent need for bioremediation, all five strains were tested for their ability to perform anaerobic denitrification with potassium nitrate and sodium nitrite as the sole nitrogen source and electron acceptor, respectively, as well as with ammonium chloride to simultaneously check for ammonia reduction.

All of the strains showed the same denitrification abilities. After 72 h no traces of nitrate or nitrite could be detected in the $KNO_3$ or $NaNO_2$ media, while at the same time $N_2$ gas was produced. The $NH_4Cl$ medium on the other hand showed no gas formation, as well as no traces of nitrate and nitrite (Table 4). These results show the ability of all of the organisms to use nitrate and nitrite for anaerobic denitrification, while ammonium could not be metabolized. It is worth noting that *P. pantotrophus* was identified as performing aerobic denitrification, a process in which the presence of $O_2$ does not inhibit the reduction of oxygenated nitrogen species [38]. This had not been tested and has the potential to be investigated in more detail in future studies. Giannopoulos et al. [17] investigated the respirome of *P. denitrificans*, a closely related strain to *P. pantotrophus*. An understanding of the respiratory network in more detail would be beneficial for assessing the applicability of these strains for environmental denitrification without the need for anoxic environments.

Further investigation into nitrogen uptake was performed by testing aerobic nitrification by all of the strains.

**Table 4.** Overview of anaerobic denitrification by the investigated strains. The cultivation was performed in airtight Hungate tubes filled with a special nutrient solution. The degradation of nitrate and nitrite, as well as the subsequent synthesis of $N_2$ gas, was used to determine the reduction of compounds. "-": no N compound detected/no gas detected; "+": N compound detected/gas detected.

| Strain | KNO₃ Medium | | | NaNO₂ Medium | | | NH₄Cl Medium | | |
|--------|-----|------------------|------------------|-----|------------------|------------------|-----|------------------|------------------|
| | Gas | $NO_2^-$ Detection | $NO_3^-$ Detection | Gas | $NO_2^-$ Detection | $NO_3^-$ Detection | Gas | $NO_2^-$ Detection | $NO_3^-$ Detection |
| DSM 2944$^T$ | + | - | - | + | + | nd | - | - | - |
| DSM 11072 | + | - | - | + | + | nd | - | - | - |
| DSM 11073 | + | - | - | + | + | nd | - | - | - |
| DSM 11104 | + | - | - | + | + | nd | - | - | - |
| DSM 65 | + | - | - | + | + | nd | - | - | - |

The T indicates the type strain.

### 3.5.3. The Type Strain Showed High Versatility in Nitrogen Source Usage

The BIOLOG microarrays tested a set of 95 common organic nitrogen sources, of which 67 (71%) showed a positive effect compared to the negative control (Table S2). The two amino acids L-cysteine and L-histidine, together with urea, showed the highest ΔAUC values. In general, most of the amino acids had a positive effect on respiration activity, aside from D-asparagine, L-tyrosine, L-arginine, D-alanine, L-leucine, and L-phenylalanine. Certain substances from other categories, such as nucleosides and amines, also had a very strong effect on respiration. These results verify the growth rates measured via a Growth Profiler, where the addition of L-histidine showed the highest growth rate by far. In contrast, L-cysteine hydrochloride only showed a low growth rate, despite causing a high ΔAUC here. The wide range of possible nitrogen sources again underlines the potential usage of *P. pantotrophus* in environmental applications.

Additionally, 282 peptides were screened in this array. The peptides consisted of two amino acids each, which are, in the following, categorized by their functional groups. In total, 16 combinations of functional groups are possible. The distribution of pairs is not even, with most investigated pairs falling into the hydrophobic–hydrophobic group as well as the hydrophobic–charged group. When related to the total amount of each pair, the highest percentage of positive ΔAUC can be seen in the uncharged–aromatic, aromatic–uncharged, and uncharged–hydrophobic groups (Table S3). This is of note, as L-histidine, the amino acid giving the highest growth rate for the type strain, falls into the charged category. Aromatic amino acids on their own had no noticeable positive effect on the growth rate, but together with uncharged amino acids, forming a dipeptide, they show the best influence on respiration.

### 3.6. Sulfur and Phosphorus Sources

It can be assumed that the wide range of possible organic electron donors of *P. pantotrophus* also applies to inorganic compounds. To assess the actual effect of two of the most important nutrient sources, phosphorus and sulfur, the utilization of different compounds was tested via growth measurements and a BIOLOG microarray.

### 3.6.1. Growth with Sulfur Compounds as Electron Donors Showed General Affinity to Sulfite for All Strains

*Paracoccus pantotrophus* was the first organism in which the so-called SOX (sulfur oxidation) system was characterized [6], using cell-free enzyme extracts to assess the substrate spectrum. Instead of using such a system, the actual influence of different reduced sulfur compounds on cell growth was to be assessed in this study. The ability of the strains to oxidize sodium thiosulfate, sodium sulfate, or sodium sulfite as electron donors with 15 mM carbonate as an autotrophic carbon source was tested. The precultures of all of the strains were grown on MSM with 20 mM of glucose. In the main experiment,

ammonium sulfate from MSM was replaced with the respective sulfur compounds (15 mM), and ammonium chloride (2 g/L) was used as a nitrogen source.

Most strains showed the fastest growth with glucose, despite no sulfur source being present (Table 5). Slower growth compared to a normal media composition due to the lack of a sulfur source can still be observed. DSM 11073 showed a slightly faster growth rate on carbonate but grows to lower cell densities. This means that this strain is likely able to source the necessary electrons for growth from another compound, either left over from the preculture or parts of the trace elements. The highest growth rate without organic carbon sources can be seen with strain DSM 11104 and sulfite, even though thiosulfate is more reduced.

**Table 5.** Growth on different sulfur compounds. The growth rates in $h^{-1}$ during growth on glucose and carbonate as positive and negative controls without any additional electron donor, respectively, were investigated. The ammonium sulfate from an MSM medium was replaced with ammonium chloride. Growth rates were determined by a Growth Profiler experiment, and an automatic algorithm was applied to the data. All values are based on biological triplicates. The standard deviation of every dataset was below 0.005, and is therefore omitted from the table.

| Substrate | DSM 2944[T] | DSM 11072 | DSM 11073 | DSM 11104 | DSM 65 |
|---|---|---|---|---|---|
| Glucose | 0.07 | 0.08 | 0.05 | 0.11 | 0.08 |
| Carbonate | 0.04 | 0.05 | 0.07 | 0.04 | 0.07 |
| Carbonate + Na-Thiosulfate | 0.01 | 0.03 | 0.01 | 0.01 | 0.01 |
| Carbonate + Na-Sulfate | 0.00 | 0.00 | 0.00 | 0.00 | 0.00 |
| Carbonate + Na-Sulfite | 0.05 | 0.00 | 0.03 | 0.08 | 0.00 |

The T indicates the type strain.

A statistical analysis showed that, while growth on thiosulfate occurred consistently between all of the strains, the growth rates were comparatively lower than with sulfite and significantly lower than during autotrophic growth without an additional sulfur source (MCP < 0.05). This is unusual, as this implies either an inhibition through the added sulfur sources or the ability of the strains to gain the necessary sulfur from other components of the medium. These results show less versatility in sulfur compound oxidation than the cell-free extracts, but give a good insight into suitable feedstocks for *P. pantotrophus* in actual industrial applications.

3.6.2. High Versatility for Usage of Different Phosphorus and Sulfur Sources by the Type Strain

The type strain DSM 2944[T] was screened via BIOLOG microarrays to test the respiration response in the presence of different phosphorus and sulfur sources. This was done in order to confirm the results which were obtained previously, as well as acquire additional information about a wider range of sulfur sources, as well as phosphorus, as the third most important nutrient.

Of the 35 tested sulfur sources, 22 (63%) resulted in a positive $\Delta$AUC in relation to the negative control (Table S4). Organic acids showed the highest values, together with thionines. Of the inorganic substances, dithiophosphate showed the highest $\Delta$AUC by far. L-cysteine again had a very positive effect on respiration, as it did as a nitrogen source.

Furthermore, 59 compounds as phosphorus sources were tested, of which 46 (78%) had a positive effect on respiration compared to the negative control (Table S5). Therefore, most tested sources have a significantly high effect on respiration, showing the wide metabolic range of *P. pantotrophus*. Cyclic monophosphates, as well as organic phosphates, are reasonable phosphorus sources.

With the aforementioned experiments, the influence of the major nutrients is covered. Aside from the nutritional requirements in a biotechnological environment, other factors have to be considered to ensure optimal growing conditions, which are explored in the following chapters.

*3.7. Further Abiotic Growth Factors Detail Growth Characteristics of the Type Strain*

To gain a more comprehensive picture of the benefits and limitations of *P. pantrophus*, aside from its metabolic capabilities, all of the strains were tested for growth at different pH values, and the type strain DSM 2944$^T$ was additionally investigated to determine the influence of other osmolytes and pH protection via compatible solutes on respiration.

3.7.1. The Type Strain Shows High Salt Tolerance and Positive Reaction to High Levels of Ethylene Glycol

The sensitivity against twelve different osmolytes (seven inorganic and five organic substances) at varying concentrations was investigated. The range of their osmotic activity, expressed through the water activity ($a_w$) of each compound, is listed in Table 6.

**Table 6.** Water activity ($a_w$) values for the tested osmolytes at 25 °C. Lower values indicate a higher deviation from ideal water behavior in saturated solutions, and therefore higher osmotic tension. N/A—no data available.

| Osmolyte | $a_w$ (25 °C) | Source |
|---|---|---|
| NaNO$_2$ | 0.60 | Apelblat [39] |
| Urea | 0.73 | Scatchard [40] |
| NaNO$_3$ | 0.74 | Apelblat [39] |
| NaCl | 0.75 | Resnik [41] |
| (NH$_4$)$_2$SO$_4$ | 0.80 | Resnik [41] |
| KCl | 0.85 | Resnik [41] |
| Na-Formate | 0.88 | Peng [42] |
| Na$_2$SO$_4$ | 0.89 | Guendouzi [43] |
| Na-Phosphate | 0.93 | Guendouzi [44] |
| Na-Lactate | 0.99 | Chen [45] |
| Na-Benzoate | N/A | |
| Ethylene glycol | N/A | |

Regarding the inorganic osmolytes, nitrate and nitrite mainly had negative effects, even in low concentrations. In contrast, most other compounds showed rather high ΔAUC values (Figure 5). Even 9% NaCl did not suppress respiration, although the $a_w$ value is comparable to that of nitrite and nitrate. KCl showed the highest ΔAUC values at 4%. All of these compounds have a toxic effect on bacterial cells [46], so it is quite remarkable that the type strain can endure such high concentrations.

In regard to organic osmolytes, ethylene glycol showed by far the most positive effect on respiration, with ΔAUC values above 80,000,000 (Figure 5). As shown before, the type strain features a μmax of 0.19 h$^{-1}$ with ethylene glycol as the sole carbon source, which shows that the wild type is able to utilize this compound. It is also known that ethylene glycol, compared with its polymeric form, shows relatively low osmotic pressure, as the molecule is small enough to pass the cell membrane without damaging it [47]. Therefore, it can be argued that ethylene glycol functions as a protectant rather than an osmolyte, much like the other organic osmolytes shown here [48].

Subsequently, additional substances known to have a protective effect on microbes during osmolytic stress, called compatible solutes, were evaluated (Table S6). For this, 6% NaCl was added to the culture broth and respiration was measured after the addition of compatible solutes. Of the 23 tested osmotic regulators, only 5 (21%) had a positive effect on respiration. Glutathione, a strong antioxidant tripeptide, showed the highest ΔAUC values, indicating that these substances might be suitable for industrial-scale fermentations with *P. pantotrophus* if protection against osmotic stresses is required.

In comparison, as can be seen in Figure 3, the most NaCl-susceptible strain was DSM 11104, as it showed the lowest respiration at 8% NaCl on a standard medium.

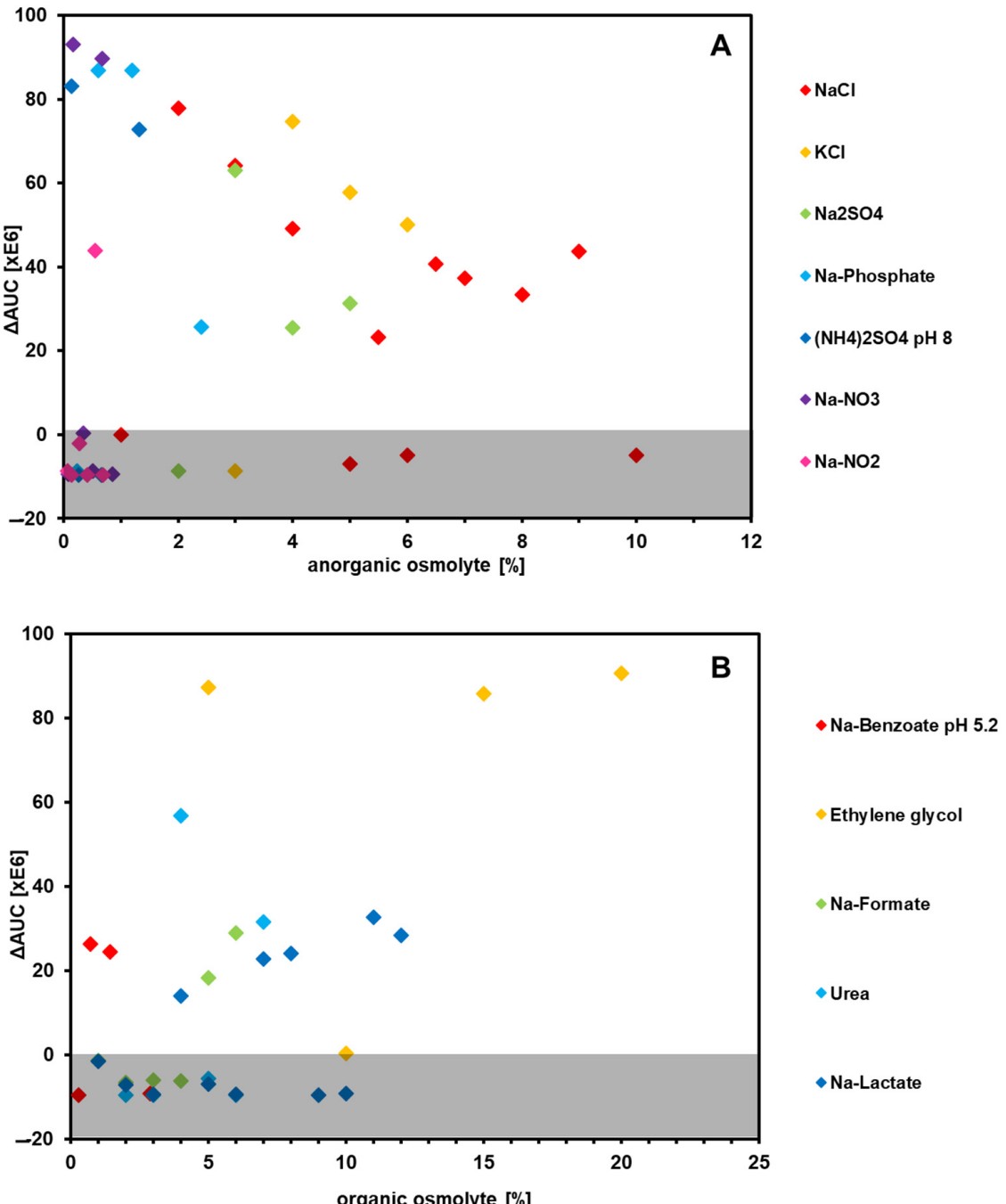

**Figure 5.** Diagrams of ΔAUC (area under a curve) values against the amount of osmolytes in percent. Shown are the values produced by inorganic salts (**A**) and organic compounds (**B**). The grey area shows all values below zero, indicating a negative effect on respiration, while all values above zero indicate a positive effect.

### 3.7.2. pH Resistance and the Influence of Compatible Solutes

To test the growth capabilities of different pH values, all five strains were examined in growth experiments with pH values ranging from 6 to 12.8 (Table 7). This range was chosen as it was previously shown that none of the strains were able to thrive at a pH below 5, while at a pH of 6 respiration was observed (Figure 3).

**Table 7.** Growth rates of different pH values in $h^{-1}$. All cultures were grown in MSM + 20 mM glucose adjusted to the respective pH. Growth rates were determined by a Growth Profiler experiment, and an automatic algorithm was applied to the online data. All values represent *n* = 3. nd = not detected.

| pH | DSM 2944[T] | DSM 11072 | DSM 11073 | DSM 11104 | DSM 65 |
|----|-------------|-----------|-----------|-----------|--------|
| 6 | nd | 0.36 ± 0.03 | 0.17 ± 0.01 | nd | 0.34 ± 0.02 |
| 7 | 0.48 ± 0.04 | 0.37 ± 0.02 | 0.35 ± 0.02 | 0.33 ± 0.02 | 0.34 ± 0.02 |
| 8 | 0.49 ± 0.03 | 0.44 ± 0.02 | 0.52 ± 0.03 | 0.53 ± 0.04 | 0.34 ± 0.01 |
| 9 | 0.46 ± 0.03 | 0.36 ± 0.02 | 0.42 ± 0.03 | 0.46 ± 0.03 | 0.42 ± 0.02 |
| 10 | 0.43 ± 0.02 | 0.32 ± 0.01 | 0.33 ± 0.01 | 0.40 ± 0.02 | 0.41 ± 0.01 |
| 11 | nd | nd | nd | nd | nd |
| 12 | nd | nd | nd | nd | nd |
| 12.8 | nd | nd | nd | nd | nd |

The T indicates the type strain.

The results showed that all of the strains feature good growth up to a pH of 10, suggesting a decent tolerance to basic growing conditions. For fermentations in which basic conditions are required, *Paracoccus* might thus be a good choice.

To assess the effectivity of different pH protectants, 36 different compounds were tested for their influence on respiration under either low pH (4.5) or high pH (9.5) conditions (Table S7). Not surprisingly, more substances had a positive effect on respiration under low pH conditions than under high pH conditions (83% compared to 42%), suggesting that there is more potential for protection against acidic conditions. L-homoserine showed the highest ΔAUC, closely followed by anthranilic acid and L-lysine at a pH of 4.5, while L-tryptophan showed the highest ΔAUC, with anthranilic acid in second position. This would make anthranilic acid a suitable additive in cultures where pH stress is of concern.

### 3.7.3. A Range of Organic Solvents Are Suitable Carbon Sources for All Tested Strains

The presence of solvents in biotechnological processes is of concern, as they can be inhibitory to the production strain if present in high enough concentrations. The solvents discussed below were chosen based on their $logP_{o/w}$ values. This parameter determines the solubility of a substance in water and an organic substance, usually octanol. It is therefore an indicator for hydrophobic/hydrophilic properties [49]. Hydrophobic substances possess a positive $logP_{o/w}$ value, while hydrophilic substances have negative $logP_{o/w}$ values. Generally, substances with $logP_{o/w}$ values < 4 are considered to be toxic to bacteria, while solvents with higher $logP_{o/w}$ values usually do not show any inhibition [50]. As shown in Table 8, all strains were able to utilize a wide array of solvents as carbon sources. Phenol and methanol were the only substances tested, which did not yield a positive result. This also confirms ethylene glycol as a suitable carbon source, as growth occurred here as well.

**Table 8.** Growth of tested strains on different solvents as the sole carbon source. An MSM medium was supplied with the solvent in comparable concentrations. Cultures were incubated for up to three days in airtight serum flasks, during which changes in OD600 were measured. Rising OD determined growth.

| Solvent | $logP_{o/w}$ | DSM 2944[T] | DSM 11072 | DSM 11073 | DSM 11104 | DSM 65 |
|---------|-------------|-------------|-----------|-----------|-----------|--------|
| Ethylene glycol | −1.36 | + | + | + | + | + |
| Methanol | −0.76 | - | - | - | - | - |
| Ethanol | −0.24 | + | + | + | + | + |
| n-Butanol | 0.8 | + | + | + | + | + |
| Phenol | 1.46 | - | - | - | - | - |
| Styrene | 2.95 | + | + | + | + | + |
| Octane | 4.5 | + | + | + | + | + |
| Decane | 5.5 | Not tested | + | + | + | + |

The T indicates the type strain.

Additionally, further solvents were tested for their toxic effect alone, using an LB medium instead of MSM (Table 9).

**Table 9.** Toxicity of different solvents. An LB medium was supplied with 5% (*v/v*) of the respective solvent. Cultures were incubated for up to three days in airtight serum flasks, during which changes in OD600 were measured. Rising OD determined growth.

| Solvent | logP$_{o/w}$ | DSM 2944$^T$ | DSM 11072 | DSM 11073 | DSM 11104 | DSM 65 |
|---|---|---|---|---|---|---|
| Ethyl acetate | 0.65 | - | - | - | - | - |
| Diethyl ether | 0.9 | - | - | - | - | - |
| Benzene | 2.0 | + | + | - | - | - |
| Toluene | 2.93 | - | - | - | - | - |
| Cyclohexane | 3.2 | - | - | - | - | - |
| n-Hexane | 3.5 | - | - | - | - | - |
| 1-Decanol | 4.57 | - | - | - | - | - |

The T indicates the type strain.

Compared to the positive control, none of the strains showed growth after 48 h of incubation, aside from the type strain and DSM 11072, which showed slight growth with benzene present. With a logP$_{o/w}$ of 2.0, benzene is considered toxic for microbes, so it is of note to know that these two strains could survive the presence of this solvent.

The results show that *P. pantotrophus* has a limited tolerance against organic solvents. As these are often used in industrial processes this is a beneficial trait which could be exploited, where other microbes would fall short.

## 4. Discussion

This study presents novel information about the species *Paracoccus pantotrophus* in order to reduce the knowledge gap that exists between it and established chassis organisms. With detailed growth characteristics of these five exemplary strains, the groundwork has been laid for characterizing chassis candidates from this genus in a more sophisticated manner, e.g., via omics technologies or bioinformatic tools.

Contrary to the type strain, which was isolated from an effluent treatment plant, strains DSM 11072, 11073, and 11104 were isolated from the carbon-disulfide-producing tree *Quercus lobato* [51], while the habitat of DSM 65 is unknown. Despite this difference in origin, a considerable overlap in the investigated parameters can be seen regarding metabolic characteristics.

Understanding growth behavior at different temperatures is important for biotechnological applications, as it ensure higher space–time yields in production processes. It could be seen that the five strains were able to grow in temperatures up to 45 °C, which makes them beneficial compared to more temperature-sensitive bacteria. The option to carry out industrial-scale fermentation at higher temperatures is beneficial, since less cooling has to be applied in high-volume set-ups with high cell densities [52].

The diauxic growth behavior of DSM 2944$^T$ is of interest, as the growth kinetics of the organism can thus be fine-tuned based on the nutrients provided. This is concurrent with other studies showing a preference of *P. pantotrophus* for organic acids and diauxic growth behavior with these kinds of substrate mixes [53]. With this, *P. pantotrophus* fermentations can be more flexible by simply exchanging the used carbon source, without changing other process parameters. This could be realized in fed-batch fermentations, as the carbon source preference could prolong production time, and therefore fitting feeding protocols can be developed [54]. The reason behind the preference for organic acids before glucose was not investigated in detail, but it can be presumed that under no substrate-limited conditions acids provide a better source for carbon and energy for biomass formation. *P. pantotrophus* is known for its versatile carbon metabolism; therefore, being able to omit the pentose phosphate pathway as well as subsequent glycolysis, and instead directly use acids in the tricarboxylic acid cycle, enables cells to grow more efficiently. Nevertheless, once the acid

is depleted no obvious lag phase is observed, and the metabolism can quickly adjust to utilizing glucose for further growth. In this sense, *P. pantotrophus* is similar to *Pseudomonas* spp., which show the same preference for organic acids, before using sugars [55].

For a circular bioeconomy, a novel chassis has to be able to use unrefined and crude feedstocks, which will provide a mixture of substrates. From our results, it can be concluded that *P. pantotrophus* would be able to use specific substrates in succession, without showing any major lag phase while switching the carbon source.

Based on the capabilities of the investigated strains, different carbon sources from agriculture or food waste can be considered [56]. Strain DSM 65 was able to utilize sunflower oil as a carbon source, which could make it a candidate for the production of sustainable products through the use of waste cooking oil [57].

The high resistance against osmotic stress effectuated by ethylene glycol, as well as the ability to use this compound as a carbon source, would make this species suitable for upcycling polyethylene terephthalate (PET), of which EG is one of the monomers [58].

Glycerol could be used as a carbon source by DSM 11073 and DSM 65, but other strains were also able to utilize it after prolonged incubation, hinting at mutational adaption. This suggests that the potential pathway for glycerol degradation may be dormant in the genome and can be activated under certain environmental conditions, especially considering that the close relative *P. denitrificans* is known to degrade glycerol [21]. Glycerol as a carbon source is interesting in regard to the upcycling of raw glycerol from biodiesel production [59]. The primary contaminant of raw glycerol is methanol, which *P. pantotrophus* cannot utilize as a carbon source despite being able to use most of the other alcohols tested. This could be explored to increase growth efficiency by genetically enabling cells to degrade several carbon sources in that waste flow, such as through adaptive laboratory evolution (ALE) experiments.

It is known that *P. pantotrophus* grows facultative autotrophic, which could be demonstrated in the sulfate utilization experiments. Growth was visible in combination with thiosulfate and sodium hydrogen carbonate as the sole carbon source. If this is regarded as a proxy for $CO_2$, the attractive possibility of the production of valuable compounds directly from $CO_2$ [60,61], with the simultaneous bioremediation of sulfate contamination, could be possible [62].

One possible product that is potentially interesting for the chemical industry of *P. pantotrophus* cultivation is PHAs. *P. pantotrophus* requires nitrogen deficiency to actively accumulate PHA granules inside the cell; however, a small amount is also produced during growth. During its growth on glucose, the homopolymer polyhydroxybutyrate is produced, which in itself is not very suitable for further applications [63]; however, through co-feeding, more interesting PHA heteropolymers can be produced [64]. These substances show more variable mechanical properties and can have a broader range of applications, with which they can replace petroleum-derived plastics. Through the careful tuning of the C/N ratio and the composition of the carbon sources (natural or tailored substrate mixes, complex side streams), polymers with varying properties can be produced with one strain [57,65,66].

If PHAs are ever to be produced as a bulk product, cheap feedstocks need to be used to compete with established petrochemical products. Without more suitable sources for energy and carbon, PHA production will not be competitive. Considering that up to 50% of the production cost can be made up by the substrate alone [67], using or developing a strain that can degrade inexpensive alternative sources efficiently is a promising approach to establishing PHA production for varying uses.

PHA production was not investigated in this study, but ongoing research shows promising results of *P. pantotrophus* DSM 11073 as a PHA production organism in lab-scale fermentations.

Depending on the feedstock in biotechnological processes, different sources of nitrogen are available to the organism in question. Flexibility can be added to the process if the usage of amino acids and other compounds as nitrogen sources is possible.

Our findings show that these strains could use protein-rich side streams instead of the ammonium sulfate of the usual minimal media. One such example could be whey, which was already shown to be a fitting substrate for PHA production in other strains [38,68].

This knowledge about nitrogen usage enables future engineers to choose feed streams with nitrogen sources, which would be most beneficial for this strain. On the other hand, the data also show which feedstocks are not suitable. It would be possible to either use defined amino acid mixtures or other protein sources, which have a high amount of histidine, as well as aromatic and uncharged amino acids such as maize, wheat, or sorghum distillers' grains, as well as jatropha meal or sugar beet leaves [69]. Just as with the crude carbon sources, these side streams used as nitrogen sources would make way for a sustainable economy, with less waste production and higher levels of exploitation of already-existing and usually underutilized resources.

*P. pantotrophus* was initially introduced as a sulfur-oxidizing organism. In this study, autotrophic growth with sulfite and to a lower extent thiosulfate was detected, but not with sulfate as an electron donor; however, the growth rate was significantly less compared to heterotrophic growth. Friedrich et al. [6] extensively describe the sulfur oxidation (SOX) enzyme complex in *P. pantotrophus*. In reconstituted (cell-free) SOX complexes, it was determined that the reaction rate with sulfite was lower than with thiosulfate, while both sulfide and elemental sulfur showed higher rates than thiosulfate. This is of interest as this does not directly translate to the in vivo experiments performed in this study. Sulfite caused a higher growth rate, of $0.08 \text{ h}^{-1}$, compared to that of thiosulfate. Theoretically, the oxidation of sulfur compounds would be most effective with more reduced molecules with lower oxidation states (i.e., thiosulfate or sulfide); however, cells grow faster with sulfite as an electron donor. Reducing sulfite to sulfate is a direct and fast reaction, which could be favorable for growth. Based on the published genome [70], there is no sulfite oxygenase annotated for the type strain, and there is a possibility that a not-yet-described enzyme is responsible for this reaction. Performing ANOVA and MCP showed a significant influence of the sulfur source on the growth rate. This will have to be considered for future applications regarding sulfur utilization.

As mentioned above, the type strain was isolated from a wastewater treatment plant, which would explain its high osmotic tolerance to a wide range of osmolytes. Even NaCl levels of 10% did not negatively affect the respiration of this strain. Other organic osmolytes also had little suppressive influence, and ethylene glycol showed a clear positive effect over the negative control. It can be assumed that *P. pantotrophus* possesses an effective osmoregulation apparatus, similar to that of other halotolerant organisms, which works for different kinds of ions [71].

Biotechnology is often less feasible than traditional chemical production, as the cost of running sterile batches makes them less competitive [72]. To circumvent this, the high-temperature tolerance [73], robustness against osmotic pressure from different salts, which typically contaminate side streams [74], and ability to be active at relatively high pH levels show the considerable potential that this organism and, by extension, this genus hold for future applications, which would not require elaborate and time-consuming setups [75].

In conclusion, the wide variety of possible carbon and nitrogen sources, as well as the tolerance to extreme environments and the ability to produce bioplastics, pave the way for the industrial application of this promising organism.

**Supplementary Materials:** The following supporting information can be downloaded at: https://www.mdpi.com/article/10.3390/applmicrobiol3010013/s1, Figure S1: Determining the temperature optimum for growth of the five *P. pantotrophus* strains; Table S1: Overview of ΔAUC (area under curve) of all positive tested carbon sources for *P. pantotrophus* DSM 2944^T; Table S2: Overview of ΔAUC (area under curve) of all positive tested nitrogen sources for *P. pantotrophus* DSM 2944^T; Table S3: Percentage of amino acid pairs which yielded a positive ΔAUC; Table S4: Overview of ΔAUC (area under curve) of all positive tested sulfur sources for *P. pantotrophus* DSM 2944^T; Table S5: Overview of ΔAUC (area under curve) of all positive tested phosphorus sources for *P. pantotrophus* DSM 2944^T; Table S6: Overview of ΔAUC (area under curve) of all tested osmotic regulators for *P. pantotrophus*

DSM 2944$^T$; Table S7: Overview of ΔAUC (area under curve) of all tested pH protective compounds for *P. pantotrophus* DSM 2944$^T$.

**Author Contributions:** D.B. analyzed the data and wrote most of the text. U.P. provided the phylogenetic analyses and the corresponding discussion. J.A.B. conducted the great majority of the practical work. L.S. carried out the BIOLOG microarrays. R.R. helped design, conduct, and analyze the data from the temperature profiling experiments. J.K. and J.B. supervised the BIOLOG and temperature experiments, respectively. L.M.B. helped conceptualize the study. T.T. evaluated the data, wrote parts of the manuscript, and conceptualized the study. All authors have read and agreed to the published version of the manuscript.

**Funding:** The authors have received funding from the Federal Ministry of Education and Research (BMBF, Germany) as part of the *ParaCoquette* project (FKZ 031B0854) and from the Excellence Initiative of the German Federal and State Governments as part of the SeedFund *PapaBio* project. The laboratory of L.M.B. is partially funded by the Deutsche Forschungsgemeinschaft (DFG, German Research Foundation) under Germany's Excellence Strategy within the Cluster of Excellence FSC 2186 "The Fuel Science Center".

**Institutional Review Board Statement:** Not applicable.

**Informed Consent Statement:** Not applicable.

**Data Availability Statement:** Not applicable.

**Acknowledgments:** The authors thank Armin Quentmeier from TU Dortmund University for valuable insights derived from decades of research with *Paracoccus*.

**Conflicts of Interest:** The authors declare no conflict of interest. The funders had no role in the design of the study; in the collection, analyses, or interpretation of data; in the writing of the manuscript; or in the decision to publish the results.

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
