# Peer review of "C-, N-, S-, and P-Substrate Spectra in and the Impact of Abiotic Factors on Assessing the Biotechnological Potential of Paracoccus pantotrophus"

_2673-8007, doi:10.3390/applmicrobiol3010013_

Round 1

Reviewer 1 Report

Interesting article that comparatively evaluates 5 strains of Paracoccus as a future model organism.

The Gram-negative alphaproteobacterium P. pantotrophus was isolated from an effluent treatment system for the removal of sulfur nitrogen compounds, originally designated with the name Thiosphaera pantotropha. Only later was this organism as signed to the genus Paracoccus. It is both facultative autotrophic and facultative anaerobic.

Due to its origin, P. pantotrophus is very well versed to survive a highly variable environment and is able to perform aerobic denitrification and heterotrophic denitrification, which enables it to anaerobic respiration instead of changing its metabolism to fer mentation. It was the first alpha proteobacterium, in which the entire sulfur oxidation (SOX) system was characterized. Furthermore, under certain nutritional limitations, 66 the production of the storage molecules polyhydroxyalkanoates (PHA) can be induced.

Figure 1. Improve the figure, consult with a graphic designer to enrich the figure, as this is irrelevant, this information can be included in the introduction of the article.

Figure 2. It cannot be reviewed in detail because it has a very low resolution. Highlight in Figure 2 the strains used in the study. Put the Bootstrap values at the nodes of the tree.

Figure 3. Organism number 5 is missing. Include details of organisms 1, 2, 3, 4, and 5 in the figure legend.

Regarding the results related to Figure 3, it is possible to put in supplementary material the overlapping genes and the detail of the core genome shared by all strains. This information is valuable and is not found in the article.

Figure 4. Improve this Figure. It is not the most representative way to present these results.

Reviewer 2 Report

General comments:

This manuscript first compares the phylogenetic status of five Paracoccus pantotrophus strains, and investigates the effect many abiotic factors on growth and metabolism, providing a relatively comprehensive introduction about P. pantotrophus strains. Lots of work have been done, and the manuscript is well written and presented. Despite a number of issues with the results and discussion that need to be addressed and enhanced, it will contribute to the future development of biotechnological strains.

Abstract:

The introduction and explanation in Lines 15-18 should be simplified since only significant results and findings should be stated. Authors only show the influence of carbon sources without other abiotic factors. Graphic abstract is suggested to be used to better display the contents of the study.

Keywords:

Not comprehensive. The chemoautotrophic and anaerobic denitrification may not be the focus of the manuscript.

Introduction:

The importance of finding a promising new microbial candidate is introduced in detail and P. pantotrophus is a potential biotechnological strain due to numerous advantages. Thus, five P. pantotrophus strains are characterized in this study via BIOLOG technology. But the innovation and importance of this study are not clear.

Materials and methods:

1.      What is the reason for conducting temperature experiment first rather than others, such as pH?

2.      The experimental conditions about carbon sources, nitrogen sources and other abiotic factors are not provided, so we do not know which carbon source and carbon concentration are chosen in the experiment of nitrogen sources.

3.      What is the formula to calculate growth rate?

Results:

1.      This part contains too much introduction of background and discussion, which should be simplified. There are too many subtitles, making the manuscript redundant and complex.

2.      Lines 229-245: the authors describe the evolutionary relationship of the five carotenoid-producing Paracoccus strains which are not used in this study. This seems to have no relevance to this paper.

3.      According to the result of section 3.2, 35-40℃ is the optimal growth temperature of P. pantotrophus, why is 30℃ selected for subsequent experiments of carbon sources? What is the meaning of temperature experiment?

4.      Although the author has done lots of experiments, each experiment seems to be independent, and the correlations between different abiotic factors are not clear, leading to weak connection of all parts.

Discussion:

1.      The authors mainly focus on the preference of different P. pantotrophus strains to carbon sources, providing some suggestions for future applications. However, the reasons that P. pantotrophus prefers organic acids over sugars are not discussed.

2.      The part of discussion needs to be comprehensive and systematic. Many abiotic factors, such as nitrogen sources, pH and so on, are investigated in this study, but relevant discussion about these are not involved.

3.      The authors discuss the possible factors that affect the production of PHA, one possible product of P. pantotrophus. However, PHA is not mentioned in result.

Table and figure:

1.      It is difficult to read the figures, especially Figures 2 and 5 due to poor quality, please improve them.

2.      The error value and significance analysis are missing in tables.

Round 2

Reviewer 2 Report

The manuscript is corrected and revised according to the reviewer's comments. I am now satisfied with the new version, so I would like to recommend its publication